# New Anti-Flavivirus Fusion Loop Human Antibodies with Zika Virus-Neutralizing Potential

**DOI:** 10.3390/ijms23147805

**Published:** 2022-07-15

**Authors:** Renato Kaylan Alves de Oliveira França, Jacyelle Medeiros Silva, Lucas Silva Rodrigues, Dimitri Sokolowskei, Marcelo Macedo Brigido, Andrea Queiroz Maranhão

**Affiliations:** 1Molecular Immunology Laboratory, Department of Cellular Biology, Institute of Biological Sciences, University of Brasilia, Brasilia 70910-900, Brazil; renatokaylan@gmail.com (R.K.A.d.O.F.); jacyelle_medeiros@hotmail.com (J.M.S.); lucassilva111898@gmail.com (L.S.R.); dimitrisokolowskei@gmail.com (D.S.); andreaqm@unb.br (A.Q.M.); 2Graduation Program in Molecular Pathology, University of Brasilia, Brasilia 70910-900, Brazil; 3Graduation Program in Molecular Biology, University of Brasilia, Brasilia 70910-900, Brazil; 4III-Immunology Investigation Institute–CNPq-MCT, São Paulo 05403-000, Brazil

**Keywords:** fusion loop, monoclonal antibody, neutralization, phage display, Zika virus

## Abstract

Zika virus infections exhibit recurrent outbreaks and can be responsible for disease complications such as congenital Zika virus syndrome. Effective therapeutic interventions are still a challenge. Antibodies can provide significant protection, although the antibody response may fail due to antibody-dependent enhancement reactions. The choice of the target antigen is a crucial part of the process to generate effective neutralizing antibodies. Human anti-Zika virus antibodies were selected by phage display technology. The antibodies were selected against a mimetic peptide based on the fusion loop region in the protein E of Zika virus, which is highly conserved among different flaviviruses. Four rounds of selection were performed using the synthetic peptide in two strategies: the first was using the acidic elution of bound phages, and the second was by applying a competing procedure. After panning, the selected VH and VL domains were determined by combining NGS and bioinformatic approaches. Three different human monoclonal antibodies were expressed as scFvs and further characterized. All showed a binding capacity to Zika (ZIKV) and showed cross-recognition with yellow fever (YFV) and dengue (DENV) viruses. Two of these antibodies, AZ1p and AZ6m, could neutralize the ZIKV infection in vitro. Due to the conservation of the fusion loop region, these new antibodies can potentially be used in therapeutic intervention against Zika virus and other flavivirus illnesses.

## 1. Introduction

Flavivirus infections, such as Zika virus disease, are a global health problem that causes relevant social and economic impacts in different countries, especially in the Americas, Africa, some European countries, and Asia, causing the death and illness of millions of people every year. Likewise, non-endemic areas are also in danger, with the potential for outbreaks due to climate change and transmission routes [1]. This scenario is complicated by the resistance found in these viruses, such as the development of mutations that provide immunological escape or even by the ADE (antibody-dependent enhancement) phenomenon that amplifies the infection. Together, these factors are prejudicial to the success of therapies and vaccines [2].

Zika virus infection outbreaks pose new challenges for clinics because of related neurological impairments, such as Guillain–Barré syndrome, meningoencephalitis, and congenital malformations, mainly due to the virus’ ability to infect neuronal progenitor cells [1,3,4]. The so-called congenital Zika virus syndrome (CZS) has become a popular object of investigation. The skeletal muscular system, the peripheral nervous system of fetuses, and the encephalon can be affected [5,6].

Zika virus is an arbovirus whose genetic material is a positive-sense single-stranded RNA molecule. The viral RNA molecule encodes a polyprotein that is cleaved into three structural proteins: capsid protein (C), pre-membrane protein (prM), and envelope protein (E), and into seven non-structural proteins (NS1, NS2A, NS2B, NS3, NS4A, NS4B, and NS5). The latter are involved in virus replication, assembly, and the inhibition of the antiviral immune response [7]. Protein E contains three domains: domain I, which represents the N-terminal portion and influences viral tropism; domain II, which comprises the dimerization region and the fusion loop; and domain III, with a binding function to membrane receptors [8,9]. The fusion loop (FL) is the most conserved E protein region among flaviviruses and plays a role in the infection process. The viral cycle begins with interactions with attachment factors on the cell surface and specific entry receptors. Then, the viral particle enters the cell mainly by clathrin-mediated endocytosis. In the late endosome, the low-pH environment promotes conformational changes in the viral envelope leading to the insertion of the fusion loop into the endosome membrane. The energy released by the interaction mediated by FL and the envelope’s structural change promotes the fusion pore’s formation, allowing for the viral genome’s release into the cytoplasm [9,10,11].

The immunotherapy of viral infections mediated by monoclonal antibodies is a relevant therapeutic approach. These molecules can block the viral infection cycle at different stages and increase antigenic presentation and cellular immune responses [12,13]. Safe and efficient neutralizing antibodies are an alternative for vaccination for emerging viruses and immunocompromised people [14,15,16,17]. Human anti-ZIKV antibodies are reported here. They were selected from a naive phage-displayed library according to their ability to bind to a ZIKV fusion loop-derived peptide. The selected VH and VL domains were combined and expressed as scFv to characterize binding and neutralizing activities. The results reveal the neutralizing potential of these antibodies against Zika virus infection.

## 2. Results

### 2.1. Antigen Design and Selection

For selecting specific antibodies to ZIKV, a peptide corresponding to the most conserved region of flavivirus, the fusion loop (FL), was designed to serve as an antigen in selecting a naive combinatorial library of antibodies expressed on the surface of phages. The antigen was designed based on the sequence alignment of the E protein domain II of different flaviviruses (Figure 1A). It was conceived to contain the FL linked, by a disulfide bridge, to a contacting loop in the domain II region, the SRCPT peptide. The use of the cysteine-bound peptide aims to bring the conformation of the peptide closer to that which occurs in the viral particle (Figure 1B).

This target peptide was also conjugated to a biotin molecule (biotinylated viral peptide) for binding to the selection plate pre-coated with streptavidin. A peptide containing only the FL amino acid sequence (free viral peptide) was also designed (Figure 1C) to serve as a competitor in the elution of specific antibodies, in addition to the usual acidic elution (Appendix A). The idea in the competitive elution was to isolate only those antibodies bound to FL, avoiding the antibodies bound to the supporting peptide SRCPT, among other unspecific antigens present in the selection platform.

A combinatorial phage-displayed library of human antibody fragments was used for selection against the biotinylated viral peptide. Antigen-specific phages were eluted with an acid solution or with a competitor peptide. Acid and competitive selection were conducted independently throughout the experiment, along with an analysis of the selection progress, to understand the antibodies’ binding and elution behavior in the system (Appendix A).

### 2.2. Selection and Identification of Enriched Variable Domains

The coding region of the VH and VL domains of antibody subpopulations from the original library (before selection) and after selection (acid or competitive elution) (Appendix A) were PCR amplified and subjected to high-performance sequencing (Illumina). The sequences of VH and VL in each round were ordered due to the enrichment of unique sequences, expressed as fold-change (FC), compared to the original library.

Sequencing statistics showed a reduction in the number of unique VH and VL sequences from round 0 (before selection) to round 4 in both phage-display strategies, indicating a successful selection process (Appendix A). The most enriched VH and VL (with the higher fold-change values) were identified. The six most enriched VHs in peptide and acid selection presented FC ranging from 483 to 1308 and 586 to 3827, respectively (Table 1). The top six enriched VLs after competitive and acidic selection showed FC varying from 80 to 2308 and 201 to 1021, respectively. The most enriched VL in the peptide eluted strategy, Lp1, had FC much higher than the other enriched VLs (Table 1).

Both selection strategies yield the same top enriched heavy chain variable domains, except for the most enriched VH, Hp1, and Hac1, which were selection-exclusive. Hac1 shows an FC of 1318 after acidic elution and Hp1 an FC of 3827 after competitive elution. Considering the top 10 selected V domains, two VH family members, IGHV1-8 and IGHV3-21, appear exclusively in the acidic selection, while IGHV1-2 is exclusive of the competitive selection (Figure 2A). Unlike VH, the top enriched VLs were more dependent on the selection procedure (Table 1). The IGKV1-33 gene family, overrepresented among enriched VLs in the competitive selection, was not selected after acidic elution. On the other hand, the IGKV4-1 gene family, the most frequent family among the VLs of acidic selection, only appears in this selection (Figure 2B). Another difference between the selections is the degree of divergence in relation to the germinal sequences in the set of VLs most enriched in each selection. (Appendix A). This degree of variation was lower in the competitive selection (*p* < 0.05).

Antigen-selected VH sequences were more divergent to germline than VL sequences (Figure 2C and Appendix A), but variable gene segments in the germline configuration were only found among VH. Hp1 and Hac1, the most enriched VHs after each elution strategy, utilized the germline configuration of IGHV1-2*02 and IGHV4-34*01 gene segments, respectively. However, Lp1, the most enriched VL after competitive selection had two amino acid changes. Other selected sequences had a high degree of sequence divergences, such as VH Hm1 and Hm2, with 24 and 19 amino acid substitutions compared to the hitherto germline, respectively (Figure 2C). This shows the enrichment of sequences with both high and low degrees of dissimilarity compared to their germinal counterpart.

### 2.3. Construction of the Recombinant Anti-Flavivirus Antibodies

Selected VH and VL domains were strategically combined in scFv (single-chain variable fragment) format, considering their enrichment, their presence or absence in both selections (prioritizing the competitive selection), and their germinal dissimilarity degree. Three scFv molecules were constructed and analyzed: AZ1p, containing Hp1; AZ3m, with Hm1; and AZ6m, with Hm2. The exceptionally enriched Lp1 was chosen as the VL counterpart of all three scFv (Figure 3). Another set of scFv molecules was constructed utilizing another VL, Lac1. However, these antibodies showed low expression levels (data not shown) and were not further analyzed.

The proposed combination of VH and VL was tested by molecular modeling and docking with the FL peptide mimetic. Three models were developed showing that the FL could bind to Fv’s paratope by inserting the Tryptophan at position 101 in a binding pocket formed by the CDRs (Figure 3C). A Histidine at position 35, conserved in the three VH, occupies the bottom of the binding pocket in close contact (cutoff of 4.5 Å) with the FL’s Tryptophan residue found buried in the paratope. The epitope interface is dominated by three residues: Trp^101^, Leu^107^, and Phe^108^, representing 37 to 45% of the total peptide interface area. The germline VL residues Tyr^32^ at CDR1 and Asp^92^ at CDR3 are spatially close, making hydrogen bonds (AZ1p e AZ6m), or stand nearby (AZ3m) the FL residues Thr^76^ and Arg^73^. A VL non-germline residue His^91^ also participates in the interface contacting either Trp^101^ (AZ1p e AZ6m) or Asn^103^ (AZ3m) (Figure 3C). Therefore, the observed model complexes supported the choice of the VH-VL combinations.

### 2.4. Production and Validation of the Recombinant Antibodies

Recombinant antibodies were produced in bacteria and purified for further characterization. The average yield of soluble scFv was 9.8 mg/liter (Appendix A). The three scFvs generated retained the ability to bind to the synthetic antigen used in the phage display selection, showing different binding profiles. AZ6m antibody showed the highest binding activity to the viral antigen, binding at least twice higher than observed with the other scFvs (Figure 4).

### 2.5. Binding to the Virus

To determine whether the generated antibodies reacted to FL in their native conformation, they were screened for binding to ZIKV, using viral particles. In contrast to peptide binding, AZ1p scFv showed the highest virus binding activity, with EC_50_ values of 14.67 ηΜ; followed by AZ6m, with EC_50_ values of 361.7 ηΜ; and finally AZ3m, with 2479.0 ηΜ (Figure 5A,D). Cross-binding to YFV and 4-DENV serotypes was also evaluated, considering the characteristic of the antigen. All antibodies showed some binding capacity to all flaviviruses tested, following the same order of binding activity observed with ZIKV (Figure 5B–D).

The binding pattern to ZIKV and YFV was similar. The differences between antibody binding to DENV serotypes were minimal and were smaller than those observed for ZIKV and YFV. AZ3m presented a very low binding capacity to ZIKV (Figure 5A). However, the antibody AZ1p, with a germinal VH, showed expressive binding to ZIKV and YFV (Figure 5A,B).

The scFvs were also tested for binding to unrelated viruses (measles virus), and no binding activity was observed, with the exception of AZ3m, which showed a weak binding at the maximum concentration tested (Appendix A).

### 2.6. Neutralizing Activity to Zika Virus Infection

A plaque reduction neutralization test (PRNT) with the Zika Virus was performed to assess whether the constructed anti-FL scFv has neutralizing capacity. The AZ1p and AZ6m recombinant antibodies showed similar neutralizing activity, with half-maximal inhibitory concentration, and IC_50_ of 397.4 ηΜ and 311.4 ηΜ, respectively (Figure 6). The AZ3m antibody did not show neutralizing capacity (IC_50_ > 100 µΜ), corroborating the low binding activity of the ZIKV particle.

## 3. Discussion

Antibodies specific to FL may have important neutralizing potential for ZIKV infection [18]. In this work, we reported the productive selection of anti-FL antibodies, reducing the library diversity throughout the rounds, warranting the selection process. The selection leads to the identification of unrelated VH and VL enriched in response to the selection. Interestingly, both elution schemes lead to a common set of VH among the top selected sequences, except for the first most enriched VH: Hp1 and Hac1. Unlike heavy chain domains, none of the most enriched VLs were selected in both schemes. The most significant differences between the selections are found in the VLs: in the FC values between the enriched VLs, the V gene families, and the degree of dissimilarity to the germinal sequences. This corroborates the idea that distinct elution schemes lead to the selection of particular antibodies and also shows that different pressure determinants drive the selection of VHs and VLs. Thus, the elution protocol impacts, at least in part, the observed selected antibodies.

Some of the selected VH and VL showed a high dissimilarity with their germline sequences. Likewise, studies of the antibody-mediated immune response to influenza virus and HIV have shown that it is possible to accumulate a vastly hypermutated VH in highly neutralizing antibodies [19,20]. However, a naive phage library is not expected to contain antigen-driven hypermutated antibodies, and it is considered a limitation for using such libraries. Even though the library used in this work may be regarded as naive for viral infection, it was assembled from donors from an endemic area for DENV. Considering that prior contact events with other flaviviruses can lead to protective immunity against ZIKV infection [21,22], preexisting highly hypermutated VH could be selected by conserved flavivirus antigens, such as those observed in AZ6m and AZ3m.

Along with hypermutated VH, the highest enrichment went to germline VHs (Hp1 and Hac1). Germline sequences are generally associated with low-affinity antibodies, but some V genes seem to possess an intrinsic antigen recognition [23]. Moreover, it has been reported that germline-like antibodies from convalescent individuals have a potent neutralizing capacity against ZIKV and SARS-CoV-2 [24,25]. Indeed, the scFv AZ1p, which contains Hp1, efficiently neutralized ZIKV. Thus, we showed that both germinal-like antibodies, such as AZ1p, and antibodies with a high number of variations, such as AZ6m, may have neutralizing potential for viral infections.

A significant challenge in developing antibodies selected with mimetic peptides is binding to the antigen in its native conformation. Moreover, the FL is a conserved viral epitope but is not easily accessible. Here, we tested the recombinant antibodies for binding to ZIKV, using whole viral particles to assess these hypotheses. From the molecular modeling analysis, it seems that the binding of all three scFvs to FL mimetic peptides involved residues of the FL, Trp^101^, Leu^107^, and Phe^108^, which are also involved in the previously reported neutralizing antibodies 2A10G6 [26] and Z6 [27]. Interestingly, the models were developed after a docking experiment, and the finding of such a similar set of contact residues supported the hypothesis that the reported scFvs could efficiently bind Zika and other similar flaviviruses. Indeed, all scFv antibodies were able to bind to ZIKV and showed up a cross-reaction to YFV and DENV. However, the binding to DENV may be underestimated. The four serotypes of DENV were tested together, which could hinder the clear differentiation of the antibodies’ binding ability to each serotype. Despite that, binding to all serotypes is expected due to the high conservation of the FL and the structural similarities between DENVs [28].

Anti-FL antibodies form a considerable part of the humoral immune response against the Zika virus, and different authors have already shown their protective efficacy in vivo with different flaviviruses [26]. The neutralizing ability of some anti-FL antibodies can be explained by the dynamism of the “viral breathing” process associated with the conformational changes induced by the antibody binding leading to the exposure of FL, even in a context without acidification [29,30]. The present work contributes two additional FL-specific ZIKV neutralizing antibodies, AZ1p and AZ6m. The neutralizing potentials observed from AZ1p (397.4 ηΜ) and AZ6m (311.4 ηΜ) are lower than some previously described anti-fusion loop antibodies such as 2A10G6 with IC_50_ = 1.67 µΜ [26,31] and C5, and IC_50_ > 10 µΜ [32] against ZIKV, but they are also higher than others anti-FL antibodies such as 3G9 (0.67 ηΜ) [33] and ZAb_FLEP (33.3 ηΜ) [34], although a direct comparison is not possible due to methodological variations that exist in the PRNT and the antibody format and valence. However, the important message from our work is that it is possible to find neutralizing antibodies in naive phage libraries, and further molecular engineering and mutagenesis may help improve FL-binding properties.

Although anti-FL antibodies are highly cross-reactive, they generally show low levels of neutralization and high levels of ADE. However, modifications in the antibody structure, such as the deletion or modification of the Fc portion, can overcome the problems associated with the ADE reaction [35]. Although in this work we used scFvs, which cannot account for ADE, these antibodies can be expressed in complete antibody formats. Additionally, in this case, their neutralizing activity must be confirmed in these new formats, and ADE activity should also be studied. In full antibody format, it is possible to use the Fc portion with mutations in its CH2 region, such as the LALA mutation [36], to avoid Fc interaction and, thus, ADE. Furthermore, the therapeutic antibody design influences its pharmacological properties. ScFv generally has a shorter half-life and a smaller molecular size. These aspects lead to a greater ability to penetrate biological tissues, such as neural tissues, and antigenic structures, such as cryptic antigens such as FL [37,38]. On the other hand, a bivalent antibody may have more significant binding and neutralization capabilities. Sharma and colleagues showed that a highly cross-reactive antibody, C10, increases its binding and neutralization capacity to ZIKV and the four types of DENV when it changes from a Fab format to a complete IgG [39]. Therefore, it is possible that the scFvs may show improvements in their capabilities when tested in a whole antibody format, with ADE activity preventing mutations.

The development of new neutralizing antibodies and the combination of antibodies with different specificities are needed to improve the efficiency of viral infection treatment, avoiding the persistence of a high viral load and the possibility of virus immune escape [40]. The successful combination of different antibodies has already been demonstrated for infection with Zika virus [40], Ebola virus [41], and HIV [42]. Since immune evasion by viral mutation was observed in non-human primates treated with anti-ZIKV antibodies with potent binding and neutralizing capacities [43,44], it is essential to dispose of a panel of antibodies targeting key conserved regions of the envelope, the main contribution of this work.

## 4. Materials and Methods

### 4.1. Cell lines and Virus Strains

Escherichia coli cells, lineage XL1-BLUE MRF’ (Stratagene, La Jolla, CA, USA, cat: 200230), with resistance to tetracycline, were grown in LB medium (Luria Bertani; 1.0% peptone from casein, 0.5% yeast extract, and 1.0% NaCl). This strain was used for the transformation, infection by phage, and DNA extraction experiments. *Escherichia coli* cells, strain Shuffle pLys Y (New England Biolabs, Ipswich, MA, USA, cat: C3030J) compatible with pET vectors (DE3), were cultured in TB medium (Terrific Broth; 2.0% Peptone from casein, 2.4% Yeast Extract, 0.4% Glycerol, 72 mM K2HPO4, 17 mM KH2PO4). This strain was used for the heterologous expression of recombinant antibodies.

Renal cells of green monkey *Cercopithecus aethiops*, Vero cells (ATCC, cat: CCL-81), were cultured in MEM medium (Eagle’s minimal essential medium; Thermo Fisher Scientific, Waltham, MA, USA, cat: 61100061). Those cells were used for plaque reduction neutralization assays (PRNT) with the recombinant antibodies (described above). Zika virus PE243 strain (GenBank MF352141) was used for ELISA assays (enzyme-linked immunosorbent assay) and neutralization assays. Yellow fever virus lineage 17DD (GenBank AF246798.1) and dengue virus type 1–4 (diagnostic kit anti-dengue type 1–4 IgM, cat: EI 266a-9601-1 M, pre-coated plates, EUROIMMUN, Lübeck, Germany) were used for the ELISA assays.

### 4.2. Preparation of the Antigens for Selection

The antigens were designed considering the protein E sequences of different flaviviruses, available in the GenBank database (DENV_1 BR/SJRP/1890/2018: MN631102.1; DENV_2 BR/SJRP/V2796/2019: MN631136.6; DENV_3 424/BR-PE/06: JX669508.1; DENV_4 Gu/SP/BR_1229: KP704217.1; WNV_H-442: AF459403.3; YFV_AR350397/Brazil/1979: U23570.1; JEV_JaOH0566: AY029207.1; ZIKV_LMM/AG5643: MT437401.1; ZIKV_PRI/PRVABC59_17/2015: MH916802.1; and ZIKV_MR766: MW143022.1) and the PDB file 5IRE. Chimera was used to explore molecular structure [45]. A mimetic peptide to the flavivirus fusion loop was designed, using the entire FL sequence, connected by a disulfide bridge to the SRCPT, a structurally associated region of domain II of the E protein of ZIKV. This peptide was synthesized conjugated to a biotin molecule (biotinylated peptide) to serve as a binding target for the fusion phages. A competitor peptide corresponding only to the fusion loop sequence was also designed. Synthetic peptides were manufactured by FastBio (Biomatik, Cambridge, ON, Canada).

### 4.3. Library Panning

Electrocompetent XL1-BLUE MRF’ cells (efficiency ~1 × 10^9^ CFU/µg) were transformed, by electroporation, with a library of M13 filamentous phage protein III-conjugated human antibodies (Fab, phagemid vector pComb3XSS), with an estimated size of 1.7 × 10^8^, generated from B cells of human peripheral blood [46]. Electroporation was performed with 0.2 cm electrical cuvettes and with the following electrical parameters: 2.5 kV, 25 µF, and 200 Ω. The transformed cells were recovered in 15 mL of SB medium (3.0% bacteriological peptone, 2.0% yeast extract, 1.0% MOPS, and pH 7.0) for 1 h at 37^o^ C and 250 rpm. For the amplification of the library, carbenicillin (resistance to the phagemid) was added to a final concentration of 20 µg/mL, and the culture was incubated for 1 h at 37 °C and 250 rpm. Then, carbenicillin was added again for a total concentration of 50 µg/mL, and the cells were cultured under the same conditions for an additional hour. The culture volume was increased to 200 mL with SB medium, and 1 mL of 1012 pfu/mL helper phage, VCSM13, glucose (1% final concentration), tetracycline (10 µg/mL final concentration, resistance to XL1-Blue cells), and carbenicillin (50 µg/mL final concentration) were added. The cells were cultured for 1.5 h at 300 rpm and 37 °C. Then, kanamycin (50 µg/mL final concentration, with resistance to the helper phage) was added, and the cells were cultured for approximately 16 h, at 300 rpm and 37 °C. After this period, the cells were centrifuged for 15 min at 4 °C and 3000 g, and the fusion phages in the supernatant were precipitated by incubation with PEG-8000 (4.0%) and NaCl (3.0%), at 4 °C for 30 min. The precipitated phages were obtained by centrifugation at 15,000 g and 4 °C for 15 min. The phages were resuspended with 2 mL of 1.0% BSA solution (bovine serum albumin, Sigma-Aldrich, St. Louis, MI, USA, cat: A7030) in TBS (50 mM Tris-HCl pH 7.5; 150 mM NaCl) and treated as input phages for selection.

The selection of Fab fusion phages was carried out using a 96-well flat-bottom NUNC PolySorp ELISA plate (Thermo Fisher Scientific, Waltham, MA, USA, cat: 456529) coated with 100 µL of 95 µM streptavidin (Thermo Fisher Scientific, Waltham, MA, USA, cat: 21122) in TBS for one hour at 37 °C. The plate was washed three times with 200 µL of TBST (TBS with 0.1% Tween-20), and 100 µL of 1.7 µM viral antigen (biotinylated peptide) in TBS was added. The plate was incubated for one hour at 37 °C and then washed. Blocking was performed with 150 µL of 3% BSA in TBST at 4 °C overnight. On the next day, 100 µL of input phages was added to the blocked plate and incubated for one hour and a half at 37 °C.

Non-binding phages were removed with a variable number of washes at each selection round (5 and 10 washes for rounds 1 and 2, respectively; 15 washes for rounds 3 and 4). The bound phages were eluted (output) through two elution strategies: one using 100 µL of 100 mM Glycine-HCl solution pH 2.2, neutralized with 6 µL of neutralizing solution (Tris-base 2 M, pH 9.1), and another using a 100 µL solution of free-peptide (unlabeled) (2.7 µM). The eluted phages were re-amplified for a new selection round, following the library amplification protocol described previously, infecting an *E. coli* culture (2 mL, at an optical density, at 600 ηm, of 1.0) with 100 µL of eluted phages for 15 min at room temperature. The infected cells were then cultured in SB medium, as described above.

### 4.4. Analysis of the Selection of Specific Antibodies

The heavy and light chain variable domains of rounds 0 and 4 (before and after both selections- acidic and competitive elution) were amplified with Platinum Taq DNA polymerase high-fidelity kit (Thermo Fisher, Waltham, MA, USA, cat: 11304029). The following oligonucleotide primers were used: sense oligonucleotide, LeadVH, and (5’-CTGCCCAACCAGCCATGGCC-3’), and antisense oligonucleotide, VH_rev, and (5’-CGATGGGCCCTTGGTGGAGGC-3’), for heavy chain; and sense oligonucleotide, Vkappa, and (5’-GGGCCCAGGCGGCCGAGCTC-3′), and antisense oligonucleotide, VKappa_rev, and (5’-AAGACAGATGGTGCAGCCACAGT-3’), for light chain. These oligonucleotides are specific for the regions of leader sequences (ompA and pelB) immediately before the V gene sequences and for the regions immediately after the J sequences of the variable domains [47]. The amplicons of VH and VL were sequenced using the Miseq system, with 2 *×* 250 bp readout from the Illumina platform. The quality of the reads was verified with the FASTQC program, and the sequencing results were analyzed with the automated immunoglobulin analysis tool, ATTILA, developed by the molecular immunology group at the University of Brasília [48]. In this tool, V(D)J signatures were determined, and the frequency variation of each sequence (fold-change) during selection was evaluated. Frequencies were calculated for each unique sequence with unique CDRs, and the fold-change (FC) value represents the changing of the frequency of each sequence at the end of selection compared to its respective frequency before selection. ATTILA also assigns a V gene segment family based on the IgBLAST sequence bank, identifying the closest germline amino acid sequence.

The degree of dissimilarity was determined as the number of variations in the amino acid sequence compared to the closest germline. The length of the amino acid CDR3 sequences of selected VH and VL were also analyzed.

### 4.5. Antibody 3D Modeling and Interaction Analysis

The modeling of the tridimensional anti-ZIKV antibody structures, using their respective VH and VL sequences, was done using RosettaAntibody [49] from ROSIE web server (https://rosie.rosettacommons.org, accessed on 17 March 2022). Only the lowest-energy structure created for each anti-ZIKV antibody was selected for further analysis. To harness the interaction between each antibody model against the targeted ZIKV fusion loop-derived peptide, global molecular docking was conducted using the ClusPro web server [50] (https://cluspro.org, accessed on 6 May 2022). Once again, the model selection criteria were the complex with the lowest energy score. The optimization and refinement of the docked-complex structures were accomplished using the SnugDock server (https://rosie.rosettacommons.org/snugdock, accessed on 13 May 2022). Finally, COCOMAPS web serve [51] (https://www.molnac.unisa.it/BioTools/cocomaps/index.psp, accessed on 15 May 2022) was chosen to visualize intermolecular contacts between antibody-peptide complexes. Chimera was used to visualize the model complexes [45]. Models were compared to PDB complexes 5JHL and 7BQ5.

### 4.6. Recombinant Antibody Design, Cloning, and Expression

Selected VH and VL were combined to construct single-chain antibody fragments (scFv). These recombinant antibodies were cloned into the pET-Sumo vector [52]. The recombinant scFv contains carboxy-terminal HIS-tag and HA-tag used for purification and detection. Antibodies were produced in Shuffle pLys Y cells, grown at 37 °C, 200 rpm, until an OD of 2.0 (600 ηm) was achieved. Induction was performed by adding 1 mM IPTG (Sigma-Aldrich, St. Louis, MI, USA, cat: I6758). Additional incubation at 200 rpm, 25 °C for 4 h was carried out. Cells were sonicated, and the clarified supernatant was purified by an immobilized metal affinity chromatography on 1 mL HISTRAP HP (Cytiva, Marlborough, MA, USA, cat: 17524701) column in AKTA system (Cytiva). The eluted fractions were cleaned and concentrated in Amicon Ultra-2 30 kDa (Merck Millipore, Burlington, MA, USA, cat: UFC203024) by diluting them in PBS (150 mM NaCl, 10 mM NaHPO4, pH 7.4) and further quantified by Bradford Assay as described in Kielkopf [53]. Antibody expression and purification were evaluated using polyacrylamide gel electrophoresis stained with Coomassie Brilliant Blue G-250 and Western-blot using a mouse anti-HA (Santa Cruz Biotechnology, Dallas, TX, USA, cat: sc-7392) at a dilution of 1:1000. Bovine anti-mouse IgG conjugated to alkaline phosphatase (Santa Cruz Biotechnology, cat: sc-2373) was used as the secondary antibody, at a dilution of 1:2000.

### 4.7. Binding of Recombinant Antibodies to Viral Peptide

One hundred µL of 240 µM recombinant antibody solution in PBS was adsorbed onto 96-well NUNC MaxiSorp ELISA plate at 4 °C overnight. The wells were washed three times with 200 µL of PBST (PBS with 0.1% Tween 20) and blocked with 150 µL of 1% BSA in PBST for 1 h at 37 °C. The wells were washed as before, and serial dilutions (3.9 µM; 1.3 µM; 0.43 µM; and 0.14 µM in 100 µL of PBS) of biotinylated viral peptide were added and incubated for 1 h at 37 °C. The plate was washed; 100 µL of alkaline phosphatase-conjugated streptavidin (SeraCare, Milford, MA, USA, cat: 475-3000), diluted 1:1000 in TBST, was added; and the plate incubated at 37 °C for 1 h. After washing the plate with PBST, the captured viral biotinylated peptides were detected with 1 mg/mL pNPP (p-nitro-phenyl-phosphate, Thermo Fisher Scientific, cat: 34045) diluted in diethanolamine substrate buffer 1× (Thermo Fisher Scientific, cat: 34064). The plate was incubated with the pNPP solution at room temperature for 30 min, and the reading was performed with a 405-nanometer filter in a SpectraMax M2e spectrophotometer (molecular devices). Mean absorbances were determined from replicates of at least two independent experiments, with antibodies from different batch and purification experiments. A mock scFv (anti-DNA), previously characterized by Dos Santos Araujo (2020) [54], produced and purified under the same conditions of the anti-ZIKV scFv, was used as a negative control.

### 4.8. Binding to Virus

Approximately 1.5 × 10^2^ PFU/mL (plaque-forming units per milliliter) of inactivated ZIKV and YFV in 100 µL of PBS (pH 7.4) was adsorbed onto a NUNC MaxiSorp ELISA plate at 4 °C overnight. The wells were washed three times with 200 µL of PBST and blocked with 150 µL of 1% BSA in PBST, for 1 h at 37 °C. Pre-coated plates with a mixture of DENV type 1–4, highly purified and in equivalent amounts (diagnostic kit EUROIMMUN, cat: EI 266a-9601-1 M, pre-coated plates), were used to analyze antibody binding to DENV. Ten three-fold serial dilutions (5.93 µM to 2.71 ηM in 100 µL of PBST) of recombinant antibodies were incubated on the virus coated plates for 1.5 h at 37°C. The plates were washed with PBST, and 100 µL of alkaline phosphatase-conjugated anti-HIS tag antibody (Sigma-Aldrich, cat: A5588), diluted 1:2000 in TBST, was added and incubated for 1 h at 37 °C. After washing the plate, bound scFvs were detected with pNPP 1 mg/mL and read at 405 nm in SpectraMax M2e spectrophotometer (molecular devices). Mean absorbances were determined from replicates of at least three independent experiments, with antibodies from different expression and purification experiments. The half effective concentration (EC_50_) of antibodies needed for half the maximal binding was determined by non-linear regression analysis in GraphPad Prism. A previously characterized [54] mock anti-DNA scFv was used as a negative control. Measles-virus-coated plates (Diagnostic kit, Enzygnost Anti-Measles virus, Siemens, Munich, Germany, cat: 429116, pre-coated plates) were also used to assess the binding of scFvs to unrelated viruses by ELISA assay as described for the flavivirus. Measles-specific polyclonal IgG was used as a positive control.

### 4.9. Virus Production and Titration

A culture flask containing 6.0 × 10^6^ Vero cells was infected for one hour at 37 °C with 0.01 MOI ZIKV in 5 mL of MEM medium with 2% FBS (serum fetal bovine; Thermo Fisher Scientific, cat: 12657029). After one hour of infection, 5 mL of MEM medium with 10% FBS and 1× antibiotic-antimycotic (Thermo Fisher Scientific, cat: 15240062) was added directly, and the culture was incubated at 37 °C and 5% CO_2_ until a cytopathic effect was observed, usually for 40 to 48 h. After this period, the culture was centrifuged at 1300× *g* at 4 °C, and the supernatant containing the virus was stored at −80 °C.

For ZIKV titration by plaque assay, 1.5 × 10^5^ Vero cells per well (24-well plate) were cultured for 24 h at 37 °C and 5% CO_2_ in 500 µL/well of MEM medium with 10% FCS and 1× antibiotic-antimycotic solution. Cells were infected with 100 µL of 10-fold dilutions of virus (10^−^^1^ to 10^−^^7^) for one hour at 37 °C. The wells were gently covered with 500 µL of MEM semi-solid medium containing 1.5% CMC (carboxymethylcellulose medium viscosity; Sigma Aldrich, cat: C4888), 2% SFB, 1× antibiotic-antimycotic, and 1% non-essential MEM amino acids (Thermo Fisher Scientific, cat: 11140068). The culture plates were incubated at 37 °C and 5% CO_2_ for four days. After this, the semi-solid medium was removed, and the cells were fixed with 500 µL of 10% formaldehyde overnight. Afterwards, the wells were washed with 1 mL of water, by incubation at 37 °C for 15 min, and then stained with 500 µL of dye solution, crystal violet 1% (Sigma-Aldrich, cat: C0775), for 30 min at room temperature. Lysis plaque counts and virus concentrations were determined in PFU/mL (plaque-forming units per ml).

### 4.10. Plaque Reduction Neutralization Assay

The neutralizing activity of the recombinant scFvs was evaluated by the test of plaque reduction neutralization (PRNT) in Vero cells. For that, 1.5 × 10^5^ cells per well were cultured for 24 h at 37 °C and 5% CO_2_ in a 24-well culture plate in 500 µL of MEM medium containing 10% FBS and 1× antibiotic-antimycotic. Eleven four-fold dilutions of the recombinant antibodies (12.2 µM to 0.01 ηM), in a volume of 100 µL of MEM medium with 2% FBS, were incubated with 100 µL of 1.24 × 10^3^ PFU/mL ZIKV (tittered for yielding 35 to 45 plates per well in a 24-well plate) for 1 h at 37 °C. The antibody and virus mixtures were added to the cells and incubated for 1 h at 37 °C. The wells were gently covered with semi-solid medium as described above. The culture plates were incubated at 37 °C and 5% CO_2_. After four days, the semi-solid medium was removed, the cells fixed and stained as described above, and the plaques were counted. The value of half of the maximum inhibitory concentration (IC_50_) corresponds to the concentration of the recombinant antibody that resulted in a reduction of half the number of plaques observed in the wells where cells were infected without the presence of antibody. IC_50_ was determined by four-parameter non-linear regression in GraphPad Prism. Neutralization values were obtained from the mean of replicates of at least two independent experiments. An anti-DNA scFv produced and purified under the same conditions of the anti-Zika scFv was used as a negative antibody. Controls were prepared with uninfected cells incubated with the highest protein concentration to assess the toxicity of the recombinant molecules.

## 5. Conclusions

We reported on two novel neutralizing recombinant antibodies, AZ1p and AZ6m, isolated from a non-immune phage display library. Their specificity and neutralizing properties support the use of a naive antibody repertoire to yield technologically relevant biomolecules. Moreover, the neutralizing activity observed for the small and monovalent scFv molecule represents an alternative for the treatment of ZIKV infection and potential interventions in infections by other flaviviruses. Future in vivo neutralization assays with ZIKV, YFV, and DENV may attest to the therapeutic potential of these antibodies.

## Figures and Tables

**Figure 1 ijms-23-07805-f001:**
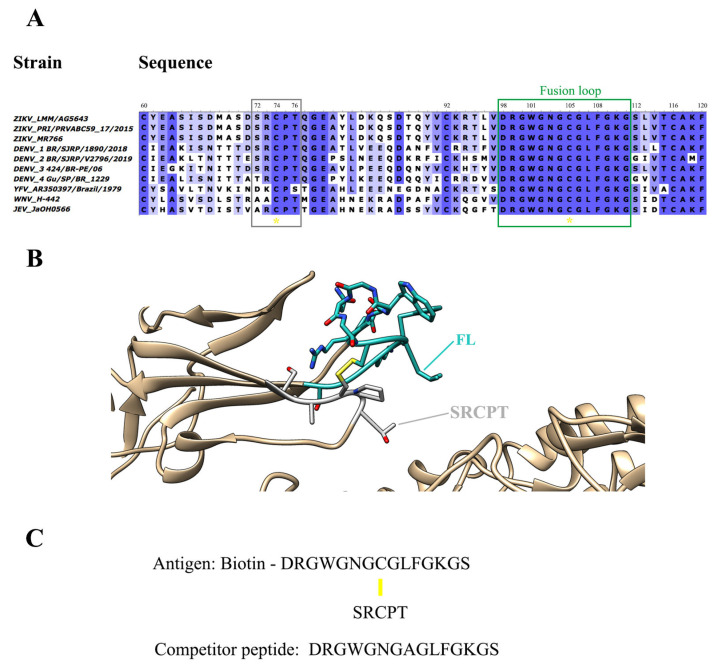
Design of antigen for the selection of antibodies anti-FL. (**A**) Antigen design was based on the alignment of the fusion loop sequences (green rectangle) found in the Protein E Domain II from different flaviviruses strains. A disulfide bridge connects the FL sequence to the SRCPT peptide (gray rectangle). Yellow asterisks highlight the conserved cysteines. (**B**) Structural representation of the FL (cyan) in the ZIKV protein E domain II. On the viral envelope, FL projects outward domain II is underpinned by the SRCPT peptide (grey) by a disulfide bridge bond (highlighted in yellow). (**C**) Peptide sequences of the selection antigen with FL, peptide SRCPT and Biotin, and competitor peptide with only FL (with an Ala substituting for Cys) are used for competitive elution. Yellow line: disulfide bridge.

**Figure 2 ijms-23-07805-f002:**
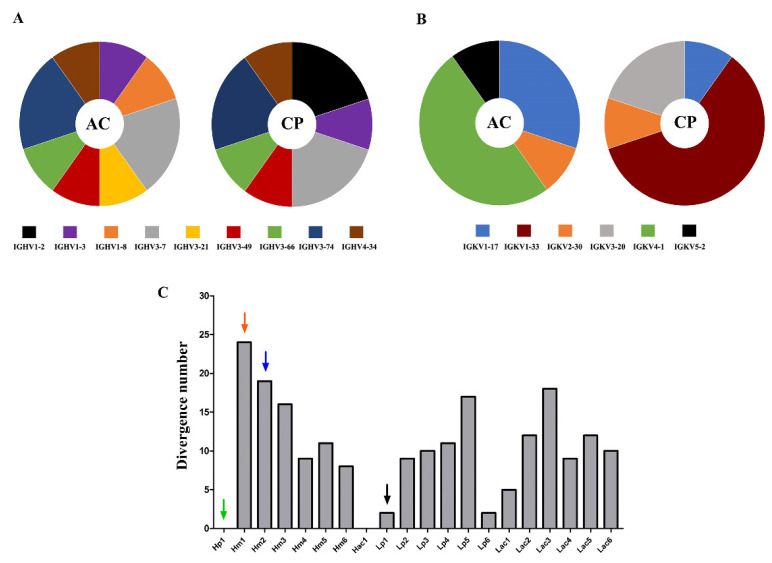
Differences between the most enriched sequences in both selection procedures. Representativeness of the V gene segment families to which the 10 most enriched VH (**A**) and VL (**B**) sequences belong, in acid (AC) and competitive selection (CP). The width of each slice of the graph is proportional to the number of sequences belonging to the specific gene family. (**C**) Analysis of the dissimilarity to germline V gene of the most enriched V domains. The degree of dissimilarity of the VH (H) and VL (L) comparing their closest germline sequences is represented by the number of non-identical residues. The arrows mark the three VH and the VL used in the tested scFv.

**Figure 3 ijms-23-07805-f003:**
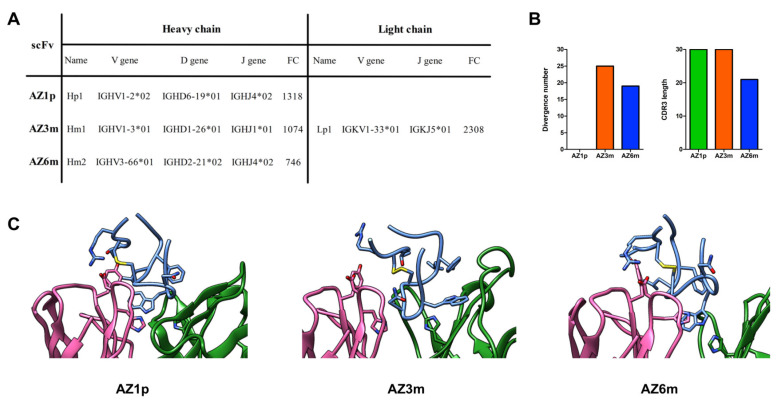
Recombinant scFvs features. Three anti-FL monoclonal antibodies, AZ1p, AZ3m, and AZ6m, were constructed by combining the enriched VHs (Hp1, Hm1, and Hm2) with the most enriched VL (Lp1), respectively, in an scFv format (**A**). Characteristics of the three scFvs, considering the variable domain of the heavy chain (**B**). Profiles of VH amino acid divergence (compared to their germline sequence) and length of the CDR3 protein sequence. (**C**) Spatial model of the paratope of the proposed scFvs. The FL peptide mimetic is in evidence in blue. VL is shown in pink and VH in blue. Amino acid residues that are discussed in the text are in evidence. The disulfide bond is marked yellow.

**Figure 4 ijms-23-07805-f004:**
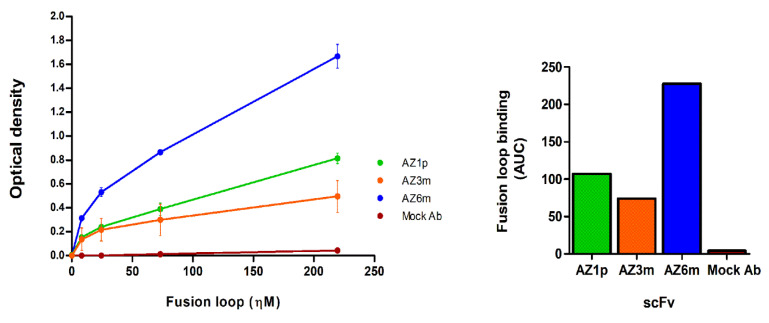
Binding of monoclonal antibodies to the fusion loop peptide mimetic. On the left, the binding of the recombinant scFvs to the fusion loop used in the selection was analyzed by immunoassay (ELISA). The normalized area under curve for each antibody FL binding is on the right. Data are presented as mean +/− SD and correspond to two independent experiments, with antibodies from different preparation batches. An unspecific scFv, produced in the same conditions as the anti-FL scFvs, was used as a negative antibody (Mock Ab).

**Figure 5 ijms-23-07805-f005:**
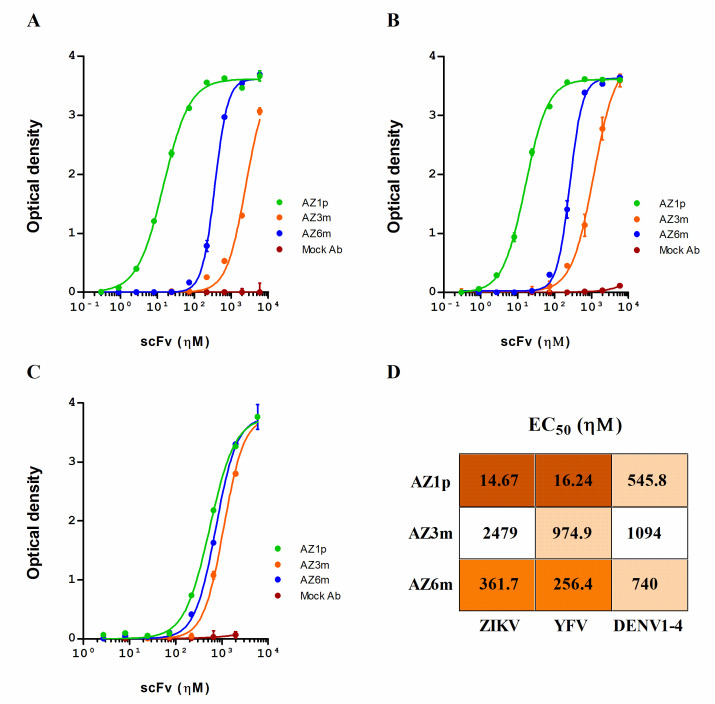
Anti-flavivirus binding activity of the selected scFvs. Recombinant scFvs (AZ1p, AZ3m, and AZ6m) were analyzed for their ability to bind to ZIKV (**A**), YFV (**B**), and an equal mix of the four DENV serotypes (**C**). In these experiments, the plates were coated with the whole viral particles. The half effective concentration (EC_50_) for each scFv against each flavivirus was also determined (**D**). Data are presented as mean +/− SD and are representative of replicates from at least three independent experiments. An unspecific scFv was used as a negative antibody (Mock Ab).

**Figure 6 ijms-23-07805-f006:**
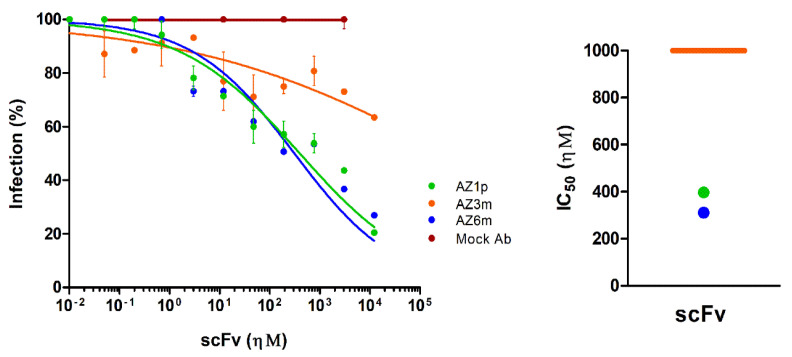
Neutralization of Zika virus infection. The ability of scFvs to neutralize the Zika virus was measured in a plaque reduction neutralization test (PRNT). On the left, the percentage of infection was calculated by the ratio between the number of plaques formed in the presence and absence of recombinant antibodies. On the right, half-maximum inhibitory concentrations (IC_50_) represent the means of replicates from at least two independent experiments. The representation of the IC_50_ values was limited to 1 µΜ. The IC_50_ value of the AZ3m greater than 1 µΜ was plotted as the limit (orange line). An unspecific scFv was used as a negative antibody (Mock Ab).

**Table 1 ijms-23-07805-t001:** Characterization of selected heavy chain variable domains (H) and light chain variable domains (L).

	Competitive Selection			Acid Selection		
	Sequence ^a^	FC ^b^	V Gene ^c^	Sequence	FC	V Gene
**VH**	Hp1	1318	IGHV1-2*02	Hac1	3827	IGHV4-34*01
	Hm1	1074	IGHV1-3*01	Hm1	1307	IGHV1-3*01
	Hm2	746	IGHV3-66*01	Hm3	838	IGHV3-7*01
	Hm3	735	IGHV3-7*01	Hm2	667	IGHV3-66*01
	Hm4	610	IGHV3-74*01	Hm6	645	IGHV3-49*04
	Hm5	483	IGHV3-7*01	Hm5	586	IGHV3-7*01
**VL**	Lp1	2308	IGHK1-33*01	Lac1	1021	IGHK1-17*03
	Lp2	395	IGHK3-20*01	Lac2	711	IGHK2-30*02
	Lp3	158	IGHK3-20*01	Lac3	679	IGHK1-17*02
	Lp4	151	IGHK2-30*02	Lac4	325	IGHK4-1*01
	Lp5	85	IGHK1-33*01	Lac5	318	IGHK1-17*01
	Lp6	80	IGHK1-33*01	Lac6	201	IGHK4-1*01

^a^ The letters “p”, “ac”, and “m” indicate a variable domain from competitive selection, acid selection, or both selections, respectively. ^b^ FC values correspond to frequency variation of a unique variable domain along with the selection procedure. ^c^ V gene family of each sequence was identified based on IgBlast annotation.

## Data Availability

Not applicable.

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
