# Peer review of "New Anti-Flavivirus Fusion Loop Human Antibodies with Zika Virus-Neutralizing Potential"

_ijms, 2022, doi:10.3390/ijms23147805_

Round 1

Reviewer 1 Report

The manuscript titled "New Anti-flavivirus Fusion Loop Human Antibodies with Zika Virus-Neutralizing Potential" discus the selection of neutralizing antibody against against a mimetic peptide based on the fusion loop region in the protein E of Zika virus, which is highly conserved among different Flavivirus. The manuscript is written and presented very well. 

Minor suggestions

1. Line 56. Need elaborate about how Fusion loop (FL) play a role in virus infection process.

2.  Line 63. Safe and efficient neutralizing antibodies are an alternative for vaccination for emerging viruses and immunocompromised people [14] More references needed.

3. Line 383-384. Method of phage selection need little more detail. e.g. which helper phage were used, how long you culture the E. coli with phage virus and helper phage. 

3. Line 380. What is the pH of neutralizing buffer (Tris-base 2M)?

Author Response

Reviewer 1: Comments and Suggestions for Authors

The manuscript titled "New Anti-flavivirus Fusion Loop Human Antibodies with Zika Virus-Neutralizing Potential" discus the selection of neutralizing antibody against against a mimetic peptide based on the fusion loop region in the protein E of Zika virus, which is highly conserved among different Flavivirus. The manuscript is written and presented very well.

Minor suggestions

  1. Line 56. Need elaborate about how Fusion loop (FL) play a role in virus infection process.

R: The reviewer's suggestion is quite pertinent. Given the construction of specific antibodies to epitopes related to the fusion loop in the article, a more detailed description of this region of the viral envelope is essential. Following the referee's suggestion, we added more information about the role of the fusion loop in the infection process in lines 55-63 to provide a better understanding of the research developed. 

Lines 55-63: The Fusion Loop (FL) is the most conserved E protein region among Flavivirus and plays a role in the infection process. The viral cycle begins with interactions with attachment factors on the cell surface and with specific entry receptors. Then, the viral particle enters the cell mainly by clathrin-mediated endocytosis. In the late endosome, the low-pH environment promotes conformational changes in the viral envelope leading to the insertion of the fusion loop into the endosome membrane. The energy released by the interaction mediated by FL along with the envelope’s structural change promotes the fusion pore's formation, allowing the viral genome's release into the cytoplasm.  [10-12].

  1. Line 63. Safe and efficient neutralizing antibodies are an alternative for vaccination for emerging viruses and immunocompromised people [14]. More references needed.

R: We are thankful for the suggestion and we added more references necessary for a better understanding of the sentence in the line 68.

  1. Line 383-384. Method of phage selection need little more detail. e.g. which helper phage were used, how long you culture the E. coli with phage virus and helper phage.

R: We are grateful for the relevant suggestion. Following this suggestion, we presented more details about the method of phage display selection in lines 362-402.

  1. Line 380. What is the pH of neutralizing buffer (Tris-base 2M)?

R: Neutralizing buffer: Tris-base 2M, pH 9.1. This information was added to line 397.

Reviewer 2 Report

The manuscript by França et al. describes about generation of monoclonal antibodies (scFvs) against Zika virus. They selected Fusion loop as a target antigen, which is a highly conserved region among flaviviruses. They have used some developed technologies, such as phage display, molecular antibody engineering etc., and have showed some characteristic features of generated antibodies. As expected, all three monoclonal antibodies could bind to ZIKV, YFV and DENVs, but only two could neutralize the live ZIKV. However, the present characterization is not sufficient to be published, authors need to add additional experiments and reveal further functions of these monoclonal antibodies.

Major comments

Figure 3: Three monoclonal antibodies are easily expected binding to FL. Authors need to determine a precise position of amino acid residue in FL recognized by each monoclonal antibody. Are there any reasons why you deduced those residues were Trp101, Phe107, and Leu108? Moreover, to my knowledge, a residue at positions 107 and 108 is Leu and Phe, respectively, but not Phe107 and Leu108.

Figure 5: Cross-reactive neutralizing activity against other flaviviruses (DENVs etc.) should be clarified by PRNT assay, since FL was intentionally chosen as a crossreactive epitope to generate monoclonal antibodies.

Line 303-304: Authors need to demonstrate that the present monoclonal antibodies have really no ADE activity by laboratory experiment using Fc gamma receptor-expressing cells. Furthermore, how long is the half-life of the present monoclonals?

Minor comment

Lines 310-311: If full-length antibodies are engineered, there is a concern about ADE. This sentence is contradictory to sentences on the lines of 300-304.

Author Response

Major comments

Figure 3: Three monoclonal antibodies are easily expected binding to FL. Authors need to determine a precise position of amino acid residue in FL recognized by each monoclonal antibody. Are there any reasons why you deduced those residues were Trp101, Phe107, and Leu108? Moreover, to my knowledge, a residue at positions 107 and 108 is Leu and Phe, respectively, but not Phe107 and Leu108.

R: The structural analysis we performed here was not a tentative for defining the precise epitope of the isolated antibodies. Instead, we used modeling and docking tools to ask if the predicted paratope formed by the isolated antibodies is compatible with an FL binding antibody. Our molecular docking simulation pointed to paratope binding to FL residues (W101, L107, and F108), also described in another FL-mAb complex published PBD structure, supporting the use of the chosen VH and VL sequences for further wet bench analysis. Binding residues were discovered using contact maps for each antibody's lowest energy docked model. Though, at this moment a detailed mapping of each antibody is beyond the scope of this research.

Indeed, there is a typo in the text, and the correct peptide residue numbering is Trp101, Leu107, and Phe108. We are in debt to the reviewer for observing it.  We correct it in the text, lines 172 and 277.

Figure 5: Cross-reactive neutralizing activity against other flaviviruses (DENVs etc.) should be clarified by PRNT assay, since FL was intentionally chosen as a crossreactive epitope to generate monoclonal antibodies.

R: We are grateful for the suggestion. However, the purpose of this article was to develop neutralizing antibodies (Ab) against zika virus infection, whose concern is increasing due to the possibility of new outbreaks, which has already been discussed in recent studies (Regla-Nava, 2022; de Matos, 2021).  

We decided to target the E protein fusion loop (FL) region instead of the domain EIII region, which has already been extensively studied for neutralizing antibodies by many authors. As this region is highly conserved, using FL as an antigen also presents the possibility of avoiding immune evasion mutations, a common phenomenon observed in antibody immunotherapies based on neutralizing antibodies.

Although we selected anti-ZIKV antibodies and tested them for binding in other flaviviruses, we do not claim that their neutralizing activity is extended beyond ZIKV. As observed in other published articles showing anti-FL antibodies in the context of a single virus (Lu, 2018; Oliphant, 2006), we focus on the neutralization of ZIKV. In the same way, it does not make it unfeasible to expand the neutralization proprieties to other flaviviruses, in future works. Moreover, neutralizing anti-FL antibodies are much less characterized than other antibodies. Thus, the work presents important neutralizing antibodies to ZIKV and it has a character of completeness for your main proposal.

-Regla-Nava JA, et al. A Zika virus mutation enhances transmission potential and confers escape from protective dengue virus immunity. Cell Rep. 2022 Apr 12;39(2):110655.

-de Matos SMS, et al. Possible Emergence of Zika Virus of African Lineage in Brazil and the Risk for New Outbreaks. Front Cell Infect Microbiol. 2021 Jul 23;11:680025.

-Lu X, et al. Double Lock of a Human Neutralizing and Protective Monoclonal Antibody Targeting the Yellow Fever Virus Envelope. Cell Rep. 2019 Jan 8;26(2):438-446.e5.

-Oliphant T, et al. Antibody recognition and neutralization determinants on domains I and II of West Nile Virus envelope protein. J Virol. 2006 Dec;80(24):12149-59.

Line 303-304: Authors need to demonstrate that the present monoclonal antibodies have really no ADE activity by laboratory experiment using Fc gamma receptor-expressing cells. Furthermore, how long is the half-life of the present monoclonals?

R: This is an interesting question. The ADE phenomenon is still a condition that needs to be further studied. Currently, it is understood only in the context of immune complexes binding (virus and antibody) to Fc receptors. Experimentally, the well-established way to assess the ADE reaction by monoclonal antibodies is in vitro infection assays with cells expressing Fc receptors, like K562 and Raji cells. These cells usually are not permissible to flaviviruses, but an ADE reaction dependent on the cell’s Fc receptors leads to an efficient virus infection. Thus, it is possible to test antibodies containing the Fc portion for inducing or not ADE reaction. Nevertheless, we understand the reviewer's questioning and we rewrote the sentences (lines 305-313) in the discussion session, so there is no misinterpretation that our antibodies do not generate ADE.

Our experiments were limited to in vitro characterization of scFv molecules. Though, at this research stage, we do not evaluate the antibodies by in vivo assays. So half-life can be addressed in future in vivo experiments.

Minor comment

Lines 310-311: If full-length antibodies are engineered, there is a concern about ADE. This sentence is contradictory to sentences on the lines of 300-304.

R: The reviewer's remark is pertinent, and we appreciate the comment. We changed lines 317-319 to correct this contradiction.